# Granulocyte Colony Stimulating Factor (GCSF) Can Attenuate Neuropathic Pain by Suppressing Monocyte Chemoattractant Protein-1 (MCP-1) Expression, through Upregulating the Early MicroRNA-122 Expression in the Dorsal Root Ganglia

**DOI:** 10.3390/cells9071669

**Published:** 2020-07-11

**Authors:** Ming-Feng Liao, Jung-Lung Hsu, Kwok-Tung Lu, Po-Kuan Chao, Mei-Yun Cheng, Hui-Ching Hsu, Ai-Lun Lo, Yun-Lin Lee, Yu-Hui Hung, Rong-Kuo Lyu, Hung-Chou Kuo, Chun-Che Chu, Long-Sun Ro

**Affiliations:** 1Department of Life Science, National Taiwan Normal University, Taipei 11677, Taiwan; mingfengliao@hotmail.com (M.-F.L.); ktlu@ntnu.edu.tw (K.-T.L.); 2Department of Neurology, Chang Gung Memorial Hospital-Linkou Medical Center and Chang Gung University College of Medicine, Taipei 33305, Taiwan; mr5894@cgmh.org.tw (M.-Y.C.); alan66625@gmail.com (A.-L.L.); esther0227@gmail.com (Y.-L.L.); namiado@gmail.com (Y.-H.H.); lyu5172@cgmh.org.tw (R.-K.L.); kuo0426@cgmh.org.tw (H.-C.K.); 1939chu@cgmh.org.tw (C.-C.C.); 3Department of Neurology, New Taipei Municipal TuCheng Hospital, Chang Gung Memorial Hospital, Linkou and Chang Gung University College of Medicine, New Taipei City 23652, Taiwan; tulu@ms36.hinet.net; 4Taipei Medical University, Graduate Institute of Humanities in Medicine and Research Center for Brain and Consciousness, Shuang Ho Hospital, Taipei 23561, Taiwan; 5Center for Neuropsychiatric Research, National Health Research Institutes, Miaoli 35053, Taiwan; chaopk@gmail.com; 6Institute of Biotechnology and Pharmaceutical Research, National Health Research Institutes, Miaoli 35053, Taiwan; 7Department of Traditional Chinese Medicine, Division of Chinese Acupuncture and Traumatology, Chang Gung Memorial Hospital-Linkou Medical Center and Chang Gung University College of Medicine, Taipei 33305, Taiwan; faithjanet@gmail.com

**Keywords:** chronic constriction injury, neuropathic pain, granulocyte colony stimulating factor, microRNA-122, monocyte chemoattractant protein-1, dorsal root ganglia, rats

## Abstract

Our previous animal studies and several human clinical trials have shown that granulocyte-colony stimulating factor (GCSF) can attenuate neuropathic pain through various mechanisms. GCSF itself is also a multipotent cytokine that can modulate microribonucleic acid (microRNA) expression profiles in vitro. In this study, we used the NanoString nCounter analysis system to screen the expression of different rodent microRNAs at early stage after nerve injury and studied the expression of related cytokines/chemokines in the dorsal root ganglia (DRGs) of rats that underwent chronic constriction injury (CCI) to explore the underlying mechanisms of the analgesic effects of GCSF. We found that microRNA-122 expression was downregulated by CCI; in contrast, GCSF treatment significantly upregulated microRNA-122 expression in the DRGs of CCI rats on the 1st day after nerve injury. We further studied the expression of different cytokines/chemokines (IL-1β, IL-6, and monocyte chemoattractant protein-1 (MCP-1)) that were modulated by microRNA-122. MCP-1 has been reported to participate in neuropathic pain development, and its expression on the DRGs of vehicle-treated CCI rats was significantly higher than that on the DRGs of sham-operated rats; in contrast, GCSF-treated rats exhibited significantly lower MCP-1 expression in the DRG than vehicle-treated rats on the 7th day after nerve injury. An early GCSF treatment can suppress MCP-1 expressions, through upregulating microRNA-122 expressions in the DRGs of CCI rats at an earlier stage, thus indirectly attenuating neuropathic pain development.

## 1. Introduction

Our previous animal studies and several similar studies have shown that granulocyte colony stimulating factor (GCSF) treatment can attenuate neuropathic pain in rats that undergo chronic constriction injury (CCI) and spinal cord injury [1,2,3,4]. Human clinical trials have also demonstrated the analgesic effects of GCSF in patients with compressive myelopathy [5,6]. In our recent studies, an early GCSF treatment was shown to recruit opioid-containing polymorphonuclear granulocytes (PMNs) to the injury site from 12 to 48 h after nerve injury, upregulate mu opioid receptor (MOR) expression on the injured nerve on days 1–3 after nerve injury, suppress proinflammatory cytokines (IL-6 and TNF-α) in the dorsal root ganglia (DRGs) on days 2–6 after nerve injury, downregulate phosphorylated p38 on day 3 after nerve injury and activated microglia on days 3–6 after nerve injury in the spinal dorsal horn, decrease IL-6 (pro-inflammatory cytokine) but increased IL-4 (anti-inflammatory cytokine) levels from days 1 to 7 in the spinal dorsal horn (SDH) after nerve injury and thus attenuate neuropathic pain development in CCI rats [1,2] (Figure 1). In contrast to our early single dose GCSF treatment, Koda et al. [3] gave GCSF treatments to CCI rats for 5 consecutive days. Mechanical allodynia of CCI rats were significantly attenuated; elevated phosphorylated p38 and IL-1β on the dorsal horn of CCI rats were significantly suppressed in their study [3]. In addition to previous anti-inflammatory effects [1,2,3,4], GCSF can also directly modulate the microRNA expression profiles of hematopoietic cells from healthy donors [7]. Microribonucleic acids (microRNAs) are small noncoding ribonucleic acid (RNA) molecules containing approximately 20–25 nucleotides and they regulate the expression of different genes post-transcriptionally by binding target genes. MicroRNAs have important roles in different research fields [8] and in chronic pain development [9,10,11,12,13,14,15]. Many studies have demonstrated temporal changes in the expression of different microRNAs in the DRGs of various animal models of inflammatory and neuropathic pain [16,17,18,19,20,21,22,23,24,25,26]. MicroRNAs can also attenuate inflammatory or neuropathic pain directly through modulating the expression of different cytokines/chemokines and ion channels in the DRGs of various animal models of pain [16,18,20,21,22,23,24,26,27,28]. In this study, we used the NanoString nCounter Analysis System [29] which can detect different gene expressions through a hybridizing special barcode system to evaluate the expression of 420 different rodent microRNAs in the DRGs of sham controls and CCI rats treated with or without GCSF at the early stage (1st day) after nerve injury. Of the 420 screened microRNAs, microRNA-122 was significantly downregulated, and early GCSF treatment upregulated microRNA-122 expression in the DRGs of CCI rats on the 1st day after nerve injury. The other 419 screened microRNAs did not show a significant temporal change. MicroRNA-122 has a specific expression in the liver and plays an important role in cholesterol metabolism [30]. Previous studies showed that microRNA-122 levels in the peripheral blood changed in the patients with drug-induced liver injury [31] and acute myocardial infarction [32]. In animal study, microRNA-122 expression decreased in the blood of rats that undergo temporary middle cerebral artery occlusion [33]. In addition, an ex vivo study showed that microRNA-122 can suppress the production of proinflammatory cytokines (IL-1β, IL-6, and MCP-1) in human hepatic stellate cells [34]. Thus, we further examined the expression of different cytokines/chemokines (IL-1β, IL-6, and monocyte chemoattractant protein-1 [MCP-1]) that were modulated by microRNA-122 [34] in the DRGs of the various treatment groups at different time points to investigate the mechanisms underlying the analgesic effect of GCSF.

## 2. Materials and Methods

### 2.1. Animals

Adult male Sprague Dawley rats (BioLASCO Taiwan Co., Ltd., Taipei, Taiwan) (weighing approximately 300–350 g) were housed in temperature-controlled (22 °C) cages; rat food pellets and water were available ad libitum. All behavioral tests were performed during the light cycle. A total of 75 rats were used in all experiments. All procedures were conducted in accordance with the Guidelines for Care and Use of Laboratory Animals and were approved by the Institutional Animal Care and Use Committee (IACUC) at the Chang Gung Memorial Hospital (IACUC: 2018120401, 24 December 2018).

### 2.2. Surgical Procedure and GCSF Treatment Protocol

Sodium pentobarbital (50 mg/kg body weight) was injected intraperitoneally (i.p.) to anesthetize the rats. Then, the rats underwent CCI surgery [35] according to the procedure describe by Bennett. Muscles on the right inguinal region were separated by forceps to expose the sciatic nerve. Using 4–0 chromic gut sutures, four ligatures were loosely tied around the proximal part of the right sciatic nerve approximately 1.0–1.5 mm apart. The ligatures only slightly reduced the nerve diameter, and epineural circulation was preserved when the ligatures were tied. Muscles on the right inguinal region were separated but without manipulation of the exposed right sciatic nerve was performed in the sham group. A single dose of GCSF (200 μg/kg, Filgrastim; Kyowa Hakko Kirin, Japan) was injected intravenously (i.v.) immediately after surgery. The same amount of normal saline was injected into the vehicle control groups. (sham, CCI with vehicle treatment, CCI with GCSF treatment on the 1st day (n = 11 in each group) and 7th day (n = 14 in each group) after surgery, respectively).

### 2.3. Behavioral Tests for Mechanical Allodynia

Behavioral studies were performed on the 1st and 7th days after surgery. Each animal was placed in a 30 × 30 × 15 cm transparent box as described in previous literature [1,2,35,36] for a 10-min habituation period before the behavioral tests to minimize stress of rats and avoid the errors of further behavioral tests. Mechanical allodynia was evaluated by von Frey hairs according to a previously described protocol [1,2]. Von Frey hairs were applied to the central region of the right plantar surface of the hind paw in the ascending order of force (0.6, 1.0, 1.4, 2.0, 4, 6, 8, 10, 15, and 26 g). When the rats showed a sharp withdrawal response or a flinch to the given filament, the bending force of that filament was defined as the mechanical threshold intensity. When a withdrawal response was established, a filament with the next lower force was used and restarted the ascending order. The final hind paw withdrawal threshold was defined as the lowest force that caused at least three withdrawals out of five consecutive applications [36]. The experimental conditions were identical for the sham-operated and experimental rats (n = 9 in each group on the 1st and 7th days after surgery, respectively).

### 2.4. MicroRNA Isolation and Purification

The rats were anesthetized with sodium pentobarbital (50 mg/kg body weight) and transcardially perfused with PBS (Sigma, St. Louis, MO, USA) on the 1st and 7th days after surgery. The right L5 and L6 DRGs were separated and collected in liquid nitrogen. MicroRNAs were extracted from the L5 and L6 DRGs by miRNeasy Mini Kits (QIAGEN, Hilden, Germany) and enriched by using the miRNeasy MinElute Cleanup kit (QIAGEN, Hilden, Germany) following the manufacturer’s instructions. The quality, integrity, and purity of the isolated RNAs were measured by a spectrophotometer (Nanodrop2000, Thermo Fisher, Waltham, MA, USA) and Bioanalyzer 2100 (Agilent Technologies, Santa Clara, CA, USA).

### 2.5. nCounter Data Analysis

The NanoString nCounter Analysis System (NanoString Technologies, Seattle, WA, USA) of the Rodent miRNA Expression Assay Kit, which contained 420 different microRNAs (Appendix A), was used to evaluate the expression of different microRNAs in the DRGs of the different treatment groups. Extracted microRNAs (18 μL samples with concentrations 40 to 65 ng/μL) were used for nCounter microRNA sample preparations according to the manufacturer’s instructions. Then each microRNA sample was analyzed by NanoString nCounter analysis system [29]. Briefly, unique nucleic acid tags for each microRNAs species were ligated to the 3′ end of each mature microRNA. Each unique nucleic acid tags are linked to special Panel Code set coded by several different sequential fluorescent molecules. Then, the specifically microRNAs were counted via hybridization with the nCounter microRNA Panel Code Set. Each barcode with special sequential fluorescence molecules was collected and counted on the nCounter Digital Analyzer (NanoString Technologies, Seattle, WA, USA) and quantified each microRNA molecule present in each sample. Basically, one barcode with special sequential fluorescence read by machine represents one special microRNA molecule. The raw data of total 420 microRNAs levels from the samples of the 1st and 7th days after CCI were normalized individually by 2-steps normalization (positive control normalization and top-100 expressed microRNAs normalization) in NanoString nSolver software 3.0 according to NanoString Technologies’ suggestions (Appendix A). The statistical differences of normalized data of different microRNAs expressions between each group were further analyzed by the default setting of NanoString nSolver software 3.0. (n = 3 and 2 in each group on the 1st and 7th days after surgery, respectively).

### 2.6. Enzyme-Linked Immunosorbent Assay (ELISA)

The protein expression levels of IL-1β, IL-6, and MCP-1 in the right L5 and L6 DRGs were determined using IL-1β, IL-6, and MCP-1 ELISA kits (Sigma-Aldrich, St. Louis, MO, USA), respectively. The samples (100 μL of dorsal root ganglion lysate containing 50 μg of total protein) were analyzed in duplicate following the manufacturer’s instructions. The absorbance was read with an ELISA reader at a wavelength of 450 nm (SpectraMax M5, Molecular Devices Corporation Sunnyvale, CA, USA). (In MCP-1/IL-1β studies (n = 4), and IL-6 studies (n = 8)).

### 2.7. Immunohistochemistry (IHC) Studies

The rats were deeply anesthetized with sodium pentobarbital (50 mg/kg body weight) and transcardially perfused with PBS (Sigma-Aldrich, St. Louis, MO, USA) followed by a fixative solution containing 4% paraformaldehyde. The right L4-L6 DRGs were resected, placed in 4% paraformaldehyde for 4 h, and transferred to 30% sucrose at 4 °C overnight. The samples were subsequently embedded in OCT compound (Tissue-Tek 4583; Sakura, Tokyo, Japan) and rapidly frozen. For immunostaining of the DRGs, every fourth section was picked from a series of consecutive DRGs (10 μm). Tissue sections of the right DRGs were obtained using a freezing microtome (CM 3050; Leica, Nussloch, Germany) and mounted on polylysine-coated slides. (n = 4 in each group on the 7th day after surgery).

For double immunofluorescence analyses, the sections were blocked with normal donkey serum (Abcam, UK) for 1 h at room temperature and then incubated overnight at 4 °C with a mixture of primary antibody solutions containing a rabbit anti-MCP-1 (monocyte chemoattractant protein-1) antibody (1:500, GeneTex, Irvine, CA, USA) and a goat anti-CGRP (calcitonin gene-related peptide) antibody (1:1000, Abcam, UK), a mouse anti-NF200 (neurofilament 200) antibody (1:1000; Sigma-Aldrich, St. Louis, MO, USA), or an anti-IB4 (isolectin B4) antibody (Alexa Fluor 488-conjugated, 1:200, Invitrogen, Waltham, MA, USA). Afterwards, the sections were incubated with secondary antibodies (Alexa Fluor 594-conjugated donkey anti-rabbit, Alexa Fluor 488-conjugated donkey anti-goat and donkey anti-mouse, 1:1000, Jackson Immuno Research Laboratories, Baltimore, PA, USA) for 1 h at room temperature. Images of the sections were acquired using a fluorescence microscope (Olympus BX51, Tokyo, Japan) connected to a digital camera and computer; tiled images were created and analyzed using MetaMorph (version 7.8; Molecular Devices, San Jose, CA, USA). The numbers of MCP-1 + CGRP-positive, MCP-1 + IB4-positive, and MCP-1 + NF200-positive neurons in the L4-L6 DRGs of the different groups were counted by an investigator who was blinded to the status of the animals manually. The numbers of positive double-stained neurons in the whole field of each slide were analyzed by MetaMorph software preliminarily and confirmed manually by the investigator.

### 2.8. Statistical Analysis

Statistical analyses were performed using Prism software (version 8; GraphPad Software, San Diego, CA, USA). Each quantitative data point was plotted as the mean ± standard error of the mean (SEM). For behavioral experiments, the Shapiro–Wilk test was used to check if the behavioral experiments data were normally distributed. Normally distributed data were analyzed by two-way repeated measures ANOVA, followed by post hoc Tukey’s test to compare the difference between each group. For microRNAs experiments, microRNAs that had statistically significantly different expressions (*p* < 0.05) between each group (sham-operated rats versus vehicle-treated rats, vehicle-treated rats versus GCSF-treated rats) were filtered out by unpaired *t*-tests (two-tailed) of the default setting of NanoString nSolver software 3.0. For other multiple comparisons, the Shapiro–Wilk test was used to check whether the ELISA, and immunohistochemistry data were normally distributed. Normally distributed data were analyzed by one-way analysis of variance (ANOVA) followed by post hoc Tukey’s test to compare the difference between each group. If the data were not normally distributed, we used the Kruskal–Wallis and post hoc Mann–Whitney rank-sum tests (two-tailed). The Bartlett test was used to check if the groups had equal variances before performing ANOVA. If the data lacked equal variance, we used the Brown–Forsythe and Welch tests. *p* values less than 0.05 were considered statistically significant.

## 3. Results

### 3.1. Single Early Systemic GCSF Treatment Alleviated Mechanical Allodynia in CCI Rats

The paw withdrawal thresholds of the vehicle-treated CCI rats were significantly lower than those of the sham controls on the 1st and 7th days after nerve injury, as determined by von Frey filaments. In contrast, the GCSF-treated CCI rats exhibited significantly attenuated mechanical allodynia compared to that of the vehicle-treated CCI rats on the 1st and 7th days after nerve injury (n = 9 in each group; ## *p* < 0.01: vehicle-treated rats compared to sham controls; ** *p* < 0.01: GCSF-treated CCI rats compared to vehicle-treated CCI rats) (Figure 2).

### 3.2. GCSF Upregulated MicroRNA-122 Expression in the DRGs of CCI Rats on the 1st Day after Nerve Injury

The levels of microRNA-7b, microRNA-19a, microRNA-122, and microRNA-598-3p were significantly decreased, but the levels of microRNA-141 were significantly increased in the DRGs of the vehicle-treated CCI rats compared to the sham controls on the 1st day after nerve injury. However, GCSF treatment only reversed the levels of microRNA-122 expression in the DRGs of CCI rats on the 1st day after nerve injury. The levels of microRNA-122 in the DRGs of the GCSF-treated CCI rats were significantly higher than those in the vehicle-treated CCI rats. In contrast, the levels of microRNA-7b, microRNA-19a, microRNA-141, and microRNA-598-3p exhibited similar expression in the DRGs of the vehicle-treated and GCSF-treated CCI rats (Appendix A). The other screened microRNAs also did not show a similar trend as that of microRNA-122 on the 1st day after nerve injury. However, there were no significant differences in microRNA-122 levels in the DRGs between the different groups on the 7th day after nerve injury (Day 1: n = 3 in each group; Day 7: n = 2 in each group; # *p* < 0.05: vehicle-treated rats compared to sham-operated rats; * *p* < 0.05: GCSF-treated CCI rats compared to vehicle-treated CCI rats, unpaired *t*-test between each group by the default setting of NanoString nSolver software 3.0) (Figure 3A,B).

### 3.3. GCSF Decreased MCP-1 Expression in the DRGs of CCI Rats on the 7th Day after Nerve Injury

MicroRNA-122 was shown to suppress proinflammatory cytokines (IL-1β, IL-6, and MCP-1) in an in vitro study of a human hepatic stellate cell line [34]. Thus, we further evaluated the cytokine/chemokine expressions in the DRGs in the different treatment groups. We found that the vehicle-treated CCI rats exhibited significantly higher MCP-1 levels in the DRGs than the sham-operated rats on the 1st and 7th days after nerve injury. In contrast, GCSF-treated rats exhibited significantly lower MCP-1 levels in the DRGs than vehicle-treated rats on the 7th day after nerve injury only (Day 1 and 7: n = 4 per group; ## *p* < 0.01: vehicle-treated rats compared to sham-operated rats; * *p* < 0.05: GCSF-treated CCI rats compared to vehicle-treated CCI rats) (Figure 4A,B).

The vehicle-treated rats also exhibited significantly higher IL-6 levels in the DRGs than sham-operated rats on the 7th day after nerve injury. There was a tendency for GCSF-treated CCI rats to exhibit lower IL-6 levels in the DRGs than vehicle-treated CCI rats on the 1st and 7th days after nerve injury. However, the differences did not reach statistical significance (Day 1 and 7: n = 4 per group; # *p* < 0.05: vehicle-treated rats compared to sham-operated rats) (Figure 5A,B).

There was a tendency for GCSF-treated CCI rats to exhibit lower IL-1β levels in the DRGs than vehicle-treated CCI rats on the 1st and 7th days after nerve injury. There was also a similar tendency for vehicle-treated CCI rats to exhibit higher IL-1β levels in the DRGs than the sham-operated rats on the 7th day after nerve injury. However, the differences did not reach statistical significance (Day 1 and 7: n = 4 per group) (Figure 6A,B).

Immunohistochemical studies revealed that many MCP-1-positive neurons in the DRGs were co-labeled with IB4 (a marker of small-diameter, C-fiber nonpeptidergic sensory neurons) and CGRP (a marker of small- to medium-diameter peptidergic sensory neurons). There were only a few MCP-1 + NF200 (a marker of large-diameter myelinated sensory neurons)-positive neurons (Figure 7A–C). There were significantly fewer MCP-1 + CGRP-positive neurons in the DRGs of the GCSF-treated rats than in the DRGs of the vehicle-treated rats on the 7th day after nerve injury (Day 7: n = 4 per group) (Figure 7E). There was a tendency for the GCSF-treated rats to exhibit fewer MCP-1+ IB4-positive neurons in the DRGs than the vehicle-treated and sham-operated rats. However, the differences did not reach statistical significance (Day 7: n = 4 per group) (Figure 7D). In contrast, there was no significant difference in the number of MCP-1 + NF200-positive neurons in the DRGs in the different treatment groups (Day 7: n = 4 per group) (Figure 7F).

## 4. Discussion

### 4.1. GCSF Upregulated MicroRNA-122 Expression in the DRGs of CCI Rats at the Early Phase after Nerve Injury

Our study showed that the levels of microRNA-122 were downregulated in the DRGs of CCI rats on the 1st day after nerve injury but were upregulated by early GCSF treatment. Many studies have investigated the temporal profiles of microRNA expression in the DRGs of CCI rats, but those studies usually focused on microRNA expression in the late stage after nerve injury and did not achieve consistent results [18,19,20,21,24,25]. Li et al. [19] screened the expression of different microRNAs in the DRGs of CCI rats by microarray analysis with quantitative reverse transcriptase PCR and found that the levels of microRNA-341 are upregulated on the 14th day after nerve injury. However, alterations in the microRNA expression were not evaluated at an early time point in their study. Other studies focused on different microRNAs and showed that the levels of microRNA-21 [20] and microRNA-146a-5p [18] are upregulated but that the levels of microRNA-96 [21], microRNA-183-5p [24], and microRNA-34a [25] are downregulated in the DRGs of CCI rats in the late stage after nerve injury (>3 days). Zhang et al. [22] and Sun et al. [23] showed that the levels of microRNA-141 and microRNA-206 in the DRGs of CCI rats decrease from the 1st day after nerve injury and are more significantly decreased on the 7th day after nerve injury. Our study emphasized that the levels of microRNA-122 were downregulated at an early time point (1st day) after nerve injury. The early time point in our study was quite different from the late time points used in previous studies that focused on the long-term alterations in microRNA expression in the DRGs after nerve injury. Recently, Brandenburger et al. [25] also used microRNA arrays to study the temporal profiles of microRNA expression in the DRGs of CCI rats. They found that there are significant differences in the expression of 49 and three different microRNAs between CCI and sham-operated rats at 4 h and 1 day after nerve injury, respectively. However, their study only focused on alterations in microRNA-34 expression in the late stage, and there was no discussion about alterations in microRNA expression in the early stage after nerve injury.

### 4.2. MicroRNA-122 Can Suppress the Pro-Inflammatory Cytokines and MCP-1 Expressions in an Ex Vivo Study

MicroRNA-122 shows a specific expression in the liver, and it plays an important role in cholesterol metabolism, hepatocellular carcinoma development, and hepatitis C virus replication [30]. Our study showed that microRNA-122 was expressed in the DRGs of CCI rats, although the levels were low. Liu et al. [33] showed that microRNA-122 expression is decreased in the blood of rats that undergo temporary middle cerebral artery occlusion but that intravenous microRNA-122 mimic treatment attenuates neurological deficits and decreases the cerebral infarct area by modulating leukocyte extravasation and atherosclerosis signaling. In addition, several studies have shown that microRNA-122 in the peripheral blood can also be a biomarker of drug-induced liver injury [31] and acute myocardial infarction [32]. From previous studies [31,32,33], we proposed that the small amount of microRNA-122 was conveyed to the DRGs by the bloodstream. Furthermore, Nakamura et al. [34] showed that microRNA-122 can suppress the production of proinflammatory cytokines, including IL-1β, IL-6, and MCP-1, in human hepatic stellate cells.

### 4.3. MCP-1 in the DRGs and SDHs Plays an Important Role to Promote Neuropathic Pain Formation

It is well known that proinflammatory cytokines and chemokines play important roles in neuropathic pain development; the expression of several proinflammatory cytokines (e.g., IL-1β, IL-6, and TNF-α) and chemokines (e.g., MCP-1 and fractalkine) is increased in the injured nerve, DRGs, and spinal dorsal horns under neuropathic pain conditions [37,38]. MCP-1 is a chemokine that regulates the migration and infiltration of monocytes/macrophages in inflammatory conditions and has an important role in neuropathic pain development [39,40,41,42,43]. MCP-1 expression in the DRGs and spinal dorsal horns of rats subjected to CCI and spinal nerve ligation (SNL) increases from the 1st to 14th days (peaking on the 3rd to 7th days) after nerve injury [39,40]. MCP-1 expression is also increased in the DRGs of rats receiving paclitaxel treatment (a chemotherapy-induced peripheral neuropathy model) and intraplantar complete Freund’s adjuvant (CFA) injection from 4 h to 14 days after drug administration [41,42]. Immunohistochemical (IHC) studies by Zhang et al. [41] and Jeon et al. [42] showed that MCP-1-positive neurons are markedly co-labeled with CGRP and IB4 but minimally with NF200. These findings suggest that MCP-1 is expressed mainly on small nociceptive neurons. In addition to these molecular and IHC studies, a function electrophysiological study showed that MCP-1 can also enhance the excitability of neurons in the DRGs dissociated from rats that underwent chronic compression of the DRGs [43].

### 4.4. GCSF Suppressed MCP-1 Expression in the DRGs to Attenuate Neuropathic Pain

Neuropathic pain can be attenuated by inhibiting proinflammatory cytokine/chemokine expression. The intrathecal administration of IL-6 and MCP-1 antibodies can also attenuate neuropathic pain development in SNL and CCI models [40,44]. Systemic GCSF treatment can downregulate elevated MCP-1 levels in the spinal cords of mice with experimental autoimmune encephalomyelitis (EAE) [45]. However, the detailed mechanisms by which GCSF downregulates MCP-1 expression were not discussed in this prior study. Our previous study had evaluated TNF-α levels on bilateral DRGs of CCI rats with GCSF treatment [1]. Only the right DRGs (injured site) of CCI rats showed a significantly increased TNF-α level compared to the contralateral (left) side and naïve group, suggesting that only the DRGs in the injured side show a significant increase of neuroinflammation. Thus, the MCP-1 expression on the right L5 and L6 DRGs were studied but left DRGs were not studied in the current study. We obtained similar results in the current study, showing that MCP-1 expression in the DRGs was increased after nerve injury but suppressed by GCSF treatment on the 7th day after nerve injury and that there were fewer MCP-1 + CGRP-positive neurons in the DRGs of GCSF-treated rats than in the DRGs of vehicle-treated rats. Although IL-1β and IL-6 expressions were not significantly altered between the different groups, there was a tendency for IL-1β and IL-6 levels in the DRGs of the GCSF-treated CCI rats to be lower than those in the DRGs of the vehicle-treated CCI rats. The findings suggested that some of the analgesic effects of GCSF occur through the suppression of MCP-1 expression in the injured DRGs, through the upregulation of microRNA-122 expression in the early stage after nerve injury.

## 5. Conclusions

In conclusion, our study showed that microRNA-122 expression was decreased in the DRGs of CCI rats; in contrast, GCSF treatment upregulated microRNA-122 expression in the DRGs in the early stage after nerve injury. The upregulation of microRNA-122 by GCSF treatment suppressed MCP-1 expression in the DRGs in the late stage after nerve injury, which further attenuated neuropathic pain development (Figure 8). Our study suggests that we may use novel drugs that can modulate microRNA expressions rather than microRNA mimics to treat neuropathic pain in the future. However, the current study has some limitations. First, only parts of microRNAs were screened. Second, we did not perform in situ hybridization of microRNA-122 in conjunction with immunohistochemistry for MCP-1. Third, we did not use microRNA-122 mimics/inhibitors to further confirm the analgesic effect of microRNA-122. Further studies of the expression of different cytokines/chemokines in the blood and DRGs of CCI rats treated with microRNA-122 mimics/inhibitors in different animal pain models are warranted to unravel how microRNA-122 can exert its analgesic effects on neuropathic pain. In the clinical aspects, studies of microRNA-122 levels in the peripheral blood of patients with different neuropathic pain such as diabetic painful neuropathy (DPN) and postherpetic neuralgia (PHN) are warranted to clarify the role of microRNA-122 (a potential biomarker) in neuropathic pain. Moreover, if the microRNA-122 mimics can attenuate mechanical allodynia in the studies of animal models, and microRNA-122 levels are downregulated in the patients with neuropathic pain, human clinical trials of microRNA-122 mimics can be initiated to confirm the analgesic effects of microRNA-122 mimics.

## Figures and Tables

**Figure 1 cells-09-01669-f001:**
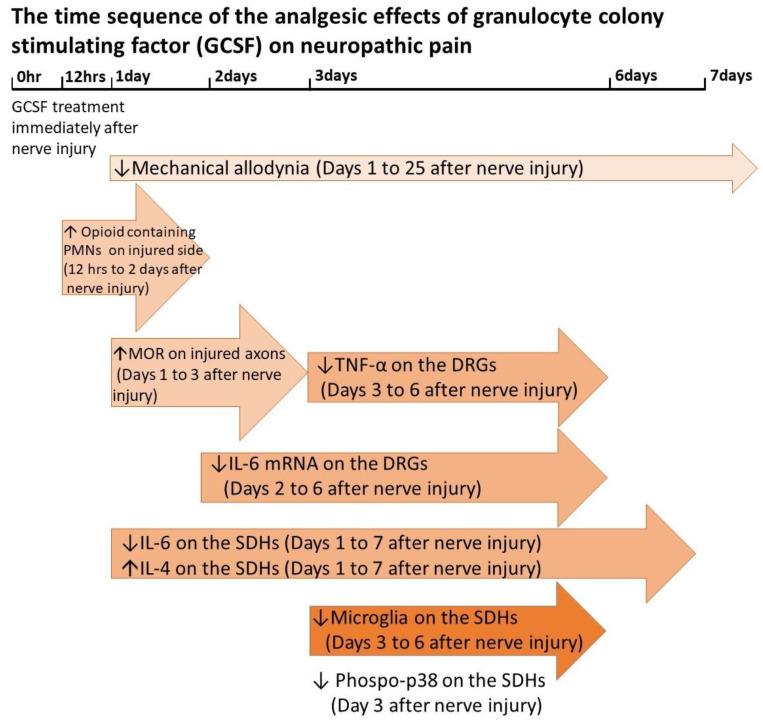
The time sequence of the analgesic effects of granulocyte colony stimulating factor (GCSF) on neuropathic pain. Compared to vehicle-treated chronic constriction injury (CCI) rats, the sequential effects of GCSF on GCSF-treated CCI rats increased opioid containing polymorphonuclear cells (PMNs) at injury site from 12 to 48 h and increased mu opioid receptor (MOR) levels on days 1–3 at the nerve ligature site after nerve injury; decreased IL-6 and TNF-α levels on days 2–6, and days 3–6, respectively, in the dorsal root ganglia (DRGs) after nerve injury; decreased IL-6 but increased IL-4 levels from days 1–7 in the spinal dorsal horns (SDHs) after nerve injury; and suppressed microglia and p-p38 activation on days 3–6 and on day 3, respectively, in the spinal dorsal horn after nerve injury [1,2]. (↑: increase, ↓: decrease).

**Figure 2 cells-09-01669-f002:**
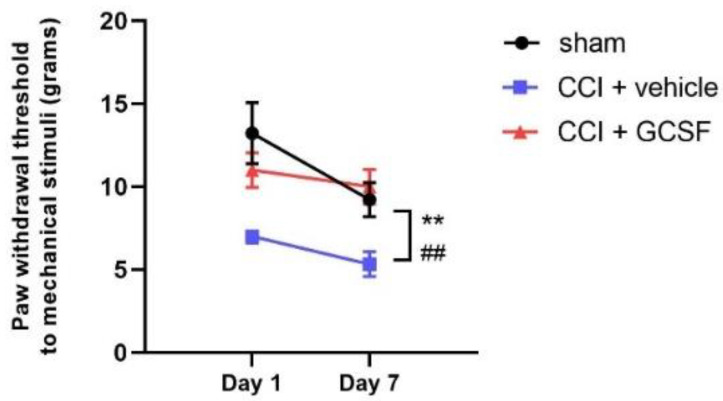
Early GCSF treatment alleviated mechanical allodynia in CCI rats on the 1st and 7th days after nerve injury. The paw withdrawal thresholds of the vehicle-treated CCI rats were significantly lower than those of the sham-operated controls on the 1st and 7th days after nerve injury, as determined by von Frey filaments (on the 1st and 7th days after nerve injury). In contrast, the GCSF-treated CCI rats exhibited significantly attenuated mechanical allodynia compared to that of the vehicle-treated CCI rats on the 1st and 7th days after nerve injury (two-way ANOVA, post hoc Tukey’s test; n = 9 in each group; ## *p* < 0.01: vehicle-treated rats compared to sham-operated controls; ** *p* < 0.01: GCSF-treated CCI rats compared to vehicle-treated CCI rats).

**Figure 3 cells-09-01669-f003:**
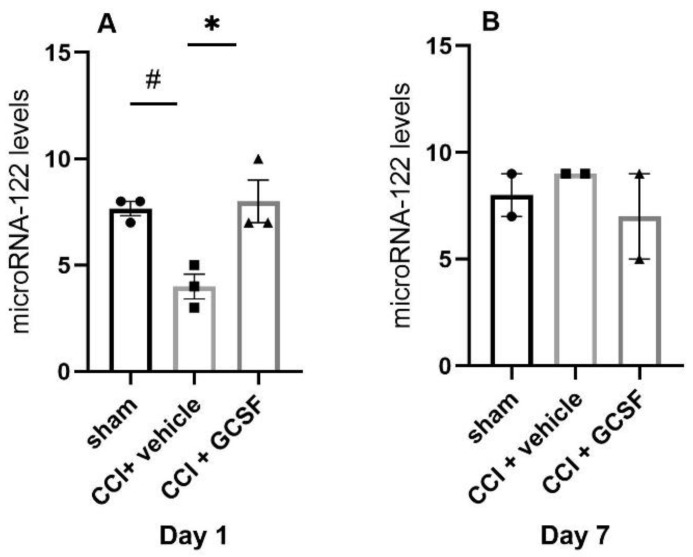
GCSF treatment upregulated the levels of microRNA-122 in the DRGs of CCI rats on the 1st day after nerve injury. MicroRNA-122 levels in the DRGs of the vehicle-treated CCI rats were significantly decreased compared to those in the DRGs of the sham-operated controls on the 1st day after nerve injury. In contrast, the microRNA levels in the DRGs of the GCSF-treated CCI rats were significantly higher than those in DRGs of the vehicle-treated CCI rats on the 1st day after nerve injury (**A**). There was no significant difference in microRNA-122 levels between sham-operated, vehicle-treated CCI, and GCSF-treated CCI rats on the 7th day after nerve injury (**B**). The other 419 screened microRNAs did not show a similar trend. The data are shown as the means ± SEMs (Day 1: n = 3 in each group; Day 7: n = 2 in each group; # *p* < 0.05: vehicle-treated rats compared to sham-operated rats; * *p* < 0.05: GCSF-treated CCI rats compared to vehicle-treated CCI rats, unpaired *t*-test between each group by the default setting of NanoString nSolver software 3.0).

**Figure 4 cells-09-01669-f004:**
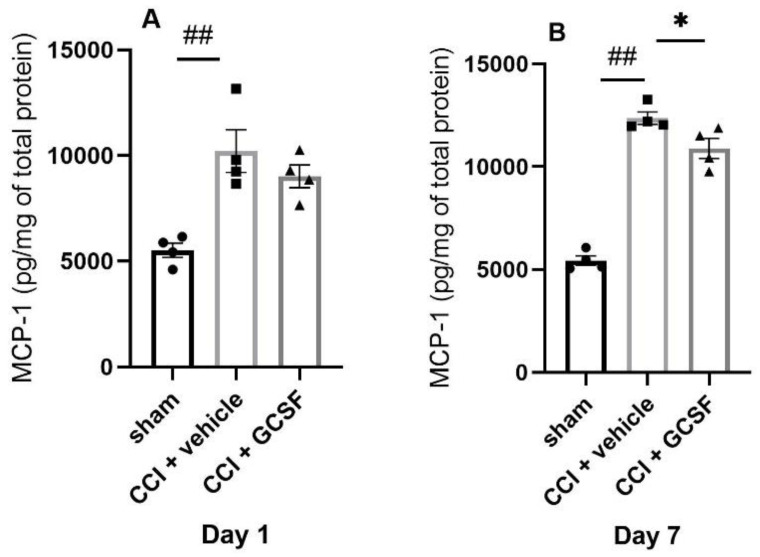
GCSF treatment decreased MCP-1 expression in the DRGs of CCI rats on the 7th day after nerve injury. The vehicle-treated CCI rats exhibited significantly higher MCP-1 levels in the DRGs than those exhibited by the sham-operated rats on the 1st and 7th days after nerve injury (**A**,**B**). In contrast, the GCSF-treated rats exhibited significantly lower MCP-1 levels in the DRGs than the vehicle-treated rats on the 7th day after nerve injury (**B**). The data are shown as the means ± SEMs (one-way ANOVA, post hoc Tukey’s test or Kruskal–Wallis, post hoc Mann–Whitney rank-sum test, if appropriate; Day 1 and 7: n = 4 per group; ## *p* < 0.01: vehicle-treated rats compared to sham-operated controls; * *p* < 0.05: GCSF-treated CCI rats compared to vehicle-treated CCI rats).

**Figure 5 cells-09-01669-f005:**
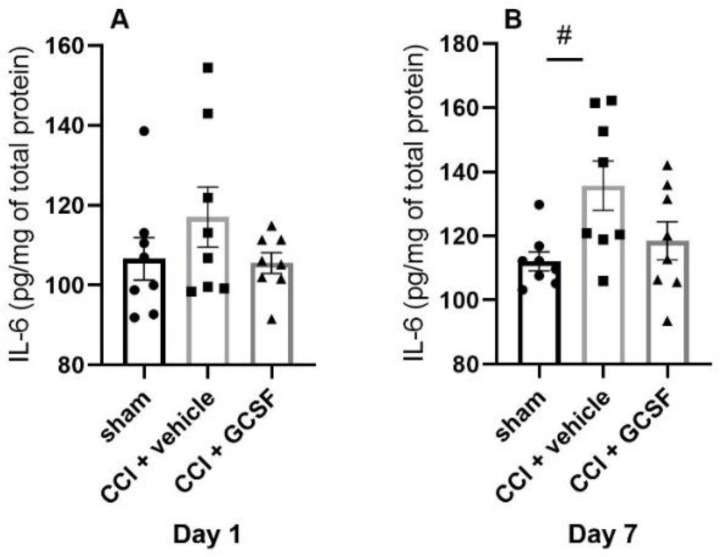
Proinflammatory cytokine IL-6 expression in the DRGs of sham-operated rats, vehicle-treated CCI rats, and GCSF-treated CCI rats. The vehicle-treated CCI rats exhibited significantly higher IL-6 levels in the DRGs than the sham-operated rats on the 7th day after nerve injury (**B**). The GCSF-treated CCI rats exhibited lower IL-6 levels in the DRGs than the vehicle-treated CCI rats on the 1st and 7th days after nerve injury (**A**,**B**). However, the difference between the groups did not reach statistical significance (one-way ANOVA, post hoc Tukey’s test or Kruskal–Wallis, post hoc Mann–Whitney rank-sum test, if appropriate; Day 1 and 7: n = 8 per group; # *p* < 0.05: vehicle-treated rats compared to sham-operated controls).

**Figure 6 cells-09-01669-f006:**
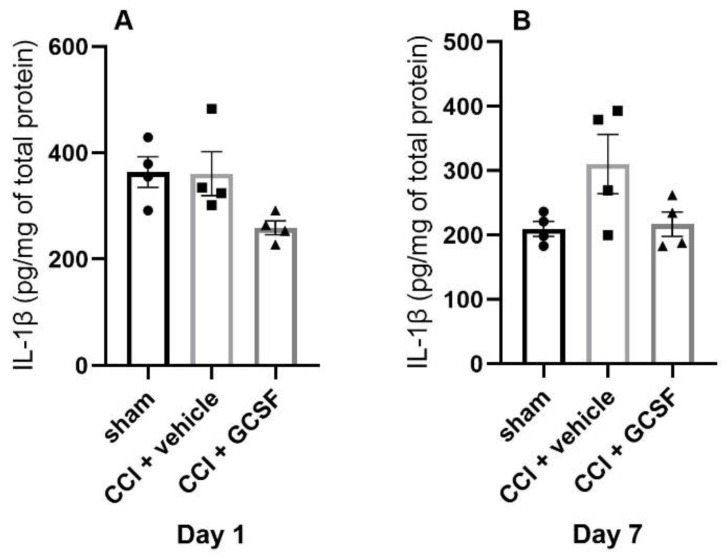
Pro-inflammatory cytokine IL-1β expression in the DRGs of sham-operated rats, vehicle-treated CCI rats, and GCSF-treated CCI rats. There was a tendency for GCSF-treated CCI rats to exhibit lower IL-1β levels in the DRGs than vehicle-treated CCI rats on the 1st and 7th days after nerve injury (**A**,**B**). There was also a tendency for vehicle-treated CCI rats to exhibit higher IL-1β levels in the DRGs than sham-operated rats on the 7th day after nerve injury (**B**). However, the differences did not reach statistical significance (one-way ANOVA, post hoc Tukey’s test or Kruskal–Wallis, post hoc Mann–Whitney rank-sum test, if appropriate; Day 1 and 7: n = 4 per group).

**Figure 7 cells-09-01669-f007:**
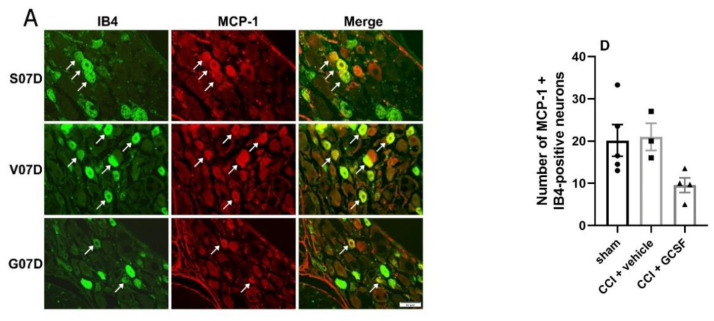
GCSF treatment decreased the number of MCP-1 + CGRP-positive neurons in the DRGs of CCI rats (**B**,**E**). Immunohistochemical studies revealed that many MCP-1-positive neurons (red) in the DRGs were co-stained with IB4 (green (**A**), a marker of small-diameter, C-fiber nonpeptidergic sensory neurons), CGRP (green (**B**), a marker of small- to medium-diameter peptidergic sensory neurons), and NF200 (green (**C**), a marker of large-diameter myelinated sensory neurons) (Figure 7A–C). Significantly fewer MCP-1 + CGRP-positive neurons were observed in the DRGs of the GCSF-treated CCI rats than in the DRGs of the vehicle-treated CCI rats on the 7th day after nerve injury (**E**). There was a tendency for the GCSF-treated CCI rats to exhibit fewer MCP-1+ IB4-positive neurons in the DRGs than the vehicle-treated CCI rats and sham-operated rats. However, the differences were not statistically significant (**D**). In contrast, there were no significant differences in the number of MCP-1 + NF200-positive neurons in the DRGs between the different groups (**F**) (one-way ANOVA, post hoc Tukey’s test or Kruskal–Wallis, post hoc Mann–Whitney rank-sum test, if appropriate; Day 7: n = 4 per group). Scale bars = 50 μm. The arrows indicate MCP-1-positive/IB4-positive, MCP-1-positive/CGRP-positive, and MCP-1-positive/NF200-positive neurons. * *p* < 0.05.

**Figure 8 cells-09-01669-f008:**
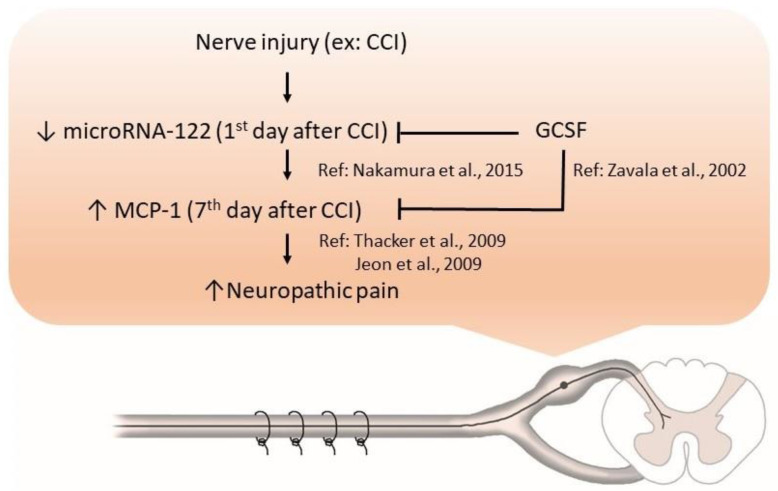
GCSF treatment downregulated MCP-1 expression in the DRGs of CCI rats, through upregulating microRNA-122 expression, which attenuated neuropathic pain. microRNA-122 expression was decreased in the DRGs of CCI rats; in contrast, GCSF treatment upregulated microRNA-122 expression in the DRGs in the early stage after nerve injury. GCSF itself directly, and the upregulation of microRNA-122 expression by GCSF treatment indirectly suppressed MCP-1 expression in the DRGs in the late stage after nerve injury, which further attenuated neuropathic pain. (CCI = chronic constriction injury, GCSF = granulocyte colony stimulating factor, MCP-1 = monocyte chemoattractant protein-1, ↑: increase, ↓: decrease.).

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
