# Peer review of "Granulocyte Colony Stimulating Factor (GCSF) Can Attenuate Neuropathic Pain by Suppressing Monocyte Chemoattractant Protein-1 (MCP-1) Expression, through Upregulating the Early MicroRNA-122 Expression in the Dorsal Root Ganglia"

_cells, 2020, doi:10.3390/cells9071669_

Round 1

Reviewer 1 Report

This preclinical study investigated the potential anti-nociceptive effect of granulocyte colony stimulating factor (GCSF) by focusing on its impact on the microRNA-122 (miRNA-122)-dependent down-regulation of monocyte chemoattactant protein-1 (MCP-1) in the dorsal root ganglion (DRG) of chronic constriction injury (CCI) rats. This study provides behavioral, biochemical, histological evidences demonstrating systemic administering injured animals with GCSF reversed the CCI-induced behavioral allodynia as well as miRNA-122 and interleukin 1 beta (IL-1b) down-regulation, MCP-1 expression and MCP-1 immunoreactivity in DRG. The author concluded that through prompting miRNA-122-suppressed MCP-1 expression in the DRG, early GCSF treatment can attenuating neuropathic pain development in CCI rats. Major comment To the first, the novelty of this study seems to be limited, for the role of GCSF-associated MCP-1/IL-1b expression/down-regulation is well-recognized; and the potential therapeutic effect of miRNAs on neuropathic pain is widely investigated. Another point is the route for drug administration. Since this study has focused on the pathology specifically in the DRG, it is puzzling to administer therapeutic reagents via an intravenous injection. Administration routes that is restricted to the target DRGs of specific segments would exclude the potential effects exhibited by GCSC other than the target region. Minor comment Though it is practical to dissect the therapeutic effect of GCSF by comparing CCI animals treated with vehicle solution and GCSF, it will be better showing the result of animal subjected to CCI for it would not only provide more clear information about the therapeutic effect of GCSF but also make sure the CCI-associated damage compared to the sham group. As shown in the RESUTS, a single bolus of GCSF did not complete reversed the behavioral allodynia, and if daily administration will exhibit more pronounced effects will be really interesting.

Author Response

Response to reviewer 1:

Major comment: To the first, the novelty of this study seems to be limited.

Thanks for your comments and suggestions. Although the potential therapeutic effect of microRNAs on neuropathic pain is widely investigated, previous studies mainly focused the changes of microRNAs expressions at the late stage after nerve injury, and were lack of detailed discussions about the microRNA expressions and interactions with proinflammatory cytokines/chemokines over time [1-6]. In contrast, our study emphasized the changes of microRNAs expressions at the early stage and studied interactions with proinflammatory cytokines/chemokines afterwards after nerve injury. These findings can develop a new strategy to treat neuropathic pain at the right time points. The “timing” is a major issue of GCSF to treat neuropathic pain.

Another point is the route for drug administration.

The current study following our previous studies discussed about the analgesic effects of GCSF, which have shown evidence of the effects of GCSF on multiple levels including the “injured nerves, ipsilateral dorsal root ganglia (DRGs) and spinal dorsal horns (SDHs) but without apparent effects on the intact nerve, contralateral DRGs and SDHs [7,8]. Moreover, Koda et al. [9] gave GCSF treatments intraperitoneally to CCI rats for 5 consecutive days. The analgesic effects of GCSF treatment in their study were similar to our results. Both studies have suggested that systemic GCSF treatments are effective on neuropathic pain. Therefore, an early single intravenous injection is more practical and convenient than a local injection to the dorsal root ganglia (DRGs) region in clinical practices. However, the reviewer 1 has pointed out an interesting question “whether daily administration of GCSF will exhibit more pronounced effects than a single bolus of GCSF or not”, we cannot answer the results right now but we may perform the studies in the future.

References:

  1. Wang, Z.; Liu, F.; Wei, M.; Qiu, Y.; Ma, C.; Shen, L.; Huang, Y. Chronic constriction injury-induced microRNA-146a-5p alleviates neuropathic pain through suppression of IRAK1/TRAF6 signaling pathway. J Neuroinflammation 2018, 15.
  2. Li, H.; Shen, L.; Ma, C.; Huang, Y. Differential expression of miRNAs in the nervous system of a rat model of bilateral sciatic nerve chronic constriction injury. Int J Mol Med 2013, 32, 219-226.
  3. Sakai, A.; Suzuki, H. Nerve injury-induced upregulation of miR-21 in the primary sensory neurons contributes to neuropathic pain in rats. Biochem Biophys Res Commun 2013, 435, 176-181.
  4. Chen, H.P.; Zhou, W.; Kang, L.M.; Yan, H.; Zhang, L.; Xu, B.H.; Cai, W.H. Intrathecal miR-96 inhibits Nav1.3 expression and alleviates neuropathic pain in rat following chronic construction injury. Neurochem Res 2014, 39, 76-83.
  5. Shi, D.N.; Yuan, Y.T.; Ye, D.; Kang, L.M.; Wen, J.; Chen, H.P. MiR-183-5p Alleviates Chronic Constriction Injury-Induced Neuropathic Pain Through Inhibition of TREK-1. Neurochem Res 2018, 43, 1143-1149.
  6. Brandenburger, T.; Johannsen, L.; Prassek, V.; Kuebart, A.; Raile, J.; Wohlfromm, S.; Kohrer, K.; Huhn, R.; Hollmann, M.W.; Hermanns, H. MiR-34a is differentially expressed in dorsal root ganglia in a rat model of chronic neuropathic pain. Neurosci Lett 2019, 708, 134365.
  7. Chao, P.K.; Lu, K.T.; Lee, Y.L.; Chen, J.C.; Wang, H.L.; Yang, Y.L.; Cheng, M.Y.; Liao, M.F.; Ro, L.S. Early systemic granulocyte-colony stimulating factor treatment attenuates neuropathic pain after peripheral nerve injury. PLoS One 2012, 7, e43680.
  8. Liao, M.F.; Yeh, S.R.; Lo, A.L.; Chao, P.K.; Lee, Y.L.; Hung, Y.H.; Lu, K.T.; Ro, L.S. An early granulocyte colony-stimulating factor treatment attenuates neuropathic pain through activation of mu opioid receptors on the injured nerve. Sci Rep 2016, 6, 25490.
  9. Koda, M.; Furuya, T.; Kato, K.; Mannoji, C.; Hashimoto, M.; Inada, T.; Kamiya, K.; Ota, M.; Maki, S.; Okawa, A., et al. Delayed granulocyte colony-stimulating factor treatment in rats attenuates mechanical allodynia induced by chronic constriction injury of the sciatic nerve. Spine (Phila Pa 1976) 2014, 39, 192-197.

Reviewer 2 Report

see file attached

Author Response

Response to Reviewer 2:

Thanks for your constructive and positive comments. We have corrected line 242, “Figure 2AB” to “Figures 3A and 3B”, corrected line 339 to “Zhang et al. [22] and Sun et al. [23]”, and corrected line 345 to “Brandenburger et al. [25]”, also elsewhere. We have added space between the word and the following references. We have also added a list of abbreviations in the legends of Figure 7 to describe what is “CCI” accordingly.

Reviewer 3 Report

An interesting manuscript in its field, I suggest the authors add a paragraph how their findings can be used in the everyday cinical practice

Author Response

Response to reviewer 3:

Thanks for your constructive and positive comments and feedback. We hope that our study can develop a new strategy to treat neuropathic pain. We have added a paragraph discuss about clinical applications of the current study on lines 410-416. “In the clinical aspects, studies of microRNA-122 levels in the peripheral blood of patients with different neuropathic pain such as diabetic painful neuropathy (DPN) and postherpetic neuralgia (PHN) are warranted to clarify the role of microRNA-122 (a potential biomarker) in neuropathic pain. Moreover, if the microRNA-122 mimics can attenuate mechanical allodynia in the studies of animal models, and microRNA-122 levels are down-regulated in the patients with neuropathic pain, human clinical trials of microRNA-122 mimics can be initiated to confirm the analgesic effects of microRNA-122 mimics”

Reviewer 4 Report

Well presented study. Would recommend eliminating the word "probably" from the title, abstract and conclusion. Please state facts or conclusions based on facts/data.

Why would upregulating microRNA-122 attenuate neuropathic pain? What is the pathophysiological connection? In Figure 7: does GCSF directly inhibit the upregulation of MCP-1 as well - please clarify.

What are the next steps for miRNA-122 as a targetable biomarker for neuropathic pain?

Author Response

Response to reviewer 4:

Thanks for your constructive and positive comments and feedback. We had eliminated the word “probably” from the title, abstract and conclusions as reviewer 4 suggested. As in our discussion, we suggested upregulating microRNA-122 can suppress expressions of MCP-1, which is a mediator promoting neuropathic pain formation [1-5]. A previous ex vivo study [6] showed that microRNA-122 can suppress the production of proinflammatory cytokines, including IL-1β, IL-6 and MCP-1 in human hepatic stellate cells. Systemic GCSF treatment can directly downregulate elevated MCP-1 levels in the spinal cords of mice with experimental autoimmune encephalomyelitis (EAE) [7]. We have added the descriptions in the legends of Figure 7 to clarify the direct effect of GCSF to inhibit MCP-1 expressions. We have also added a paragraph to discuss about future clinical-applications of microRNA-122 as a targetable biomarker for neuropathic pain on lines 410-416. In the clinical aspects, studies of microRNA-122 levels in the peripheral blood of patients with different neuropathic pain such as diabetic painful neuropathy (DPN) and postherpetic neuralgia (PHN) are warranted to clarify the role of microRNA-122 (a potential biomarker) in neuropathic pain.”

References:

  1. Jeon, S.M.; Lee, K.M.; Cho, H.J. Expression of monocyte chemoattractant protein-1 in rat dorsal root ganglia and spinal cord in experimental models of neuropathic pain. Brain Res 2009, 1251, 103-111.
  2. Thacker, M.A.; Clark, A.K.; Bishop, T.; Grist, J.; Yip, P.K.; Moon, L.D.; Thompson, S.W.; Marchand, F.; McMahon, S.B. CCL2 is a key mediator of microglia activation in neuropathic pain states. Eur J Pain 2009, 13, 263-272.
  3. Zhang, H.; Boyette-Davis, J.A.; Kosturakis, A.K.; Li, Y.; Yoon, S.Y.; Walters, E.T.; Dougherty, P.M. Induction of monocyte chemoattractant protein-1 (MCP-1) and its receptor CCR2 in primary sensory neurons contributes to paclitaxel-induced peripheral neuropathy. J Pain 2013, 14, 1031-1044.
  4. Jeon, S.M.; Lee, K.M.; Park, E.S.; Jeon, Y.H.; Cho, H.J. Monocyte chemoattractant protein-1 immunoreactivity in sensory ganglia and hindpaw after adjuvant injection. Neuroreport 2008, 19, 183-186.
  5. Sun, J.H.; Yang, B.; Donnelly, D.F.; Ma, C.; LaMotte, R.H. MCP-1 enhances excitability of nociceptive neurons in chronically compressed dorsal root ganglia. J Neurophysiol 2006, 96, 2189-2199.
  6. Nakamura, M.; Kanda, T.; Sasaki, R.; Haga, Y.; Jiang, X.; Wu, S.; Nakamoto, S.; Yokosuka, O. MicroRNA-122 Inhibits the Production of Inflammatory Cytokines by Targeting the PKR Activator PACT in Human Hepatic Stellate Cells. PLoS One 2015, 10, e0144295.
  7. Zavala, F.; Abad, S.; Ezine, S.; Taupin, V.; Masson, A.; Bach, J.F. G-CSF therapy of ongoing experimental allergic encephalomyelitis via chemokine- and cytokine-based immune deviation. J Immunol 2002, 168, 2011-2019.

Reviewer 5 Report

The authors used the NanoString nCounter analysis system to screen the expression of different rodent microRNAs at early stage after nerve injury (day 1 and 7) and studied the expression of related cytokines/chemokines in the dorsal root ganglia (DRGs) of rats with chronic constriction injury (CCI) evaluating possible underlying mechanisms of the analgesic effects of GCSF. They found that microRNA-122 expression in DRGs was downregulated by CCI; in contrast, GCSF treatment in CCI rats significantly upregulated microRNA-122 expression in the related DRGs on the 1st day after nerve injury. Also, they further studied the expression of IL-1β, IL-6, and MCP-1 that were modulated by microRNA-122 as already shown by others. The MCP-1 expression in the DRGs of vehicle-treated CCI rats was significantly higher than in the DRGs of sham-operated rats; in contrast, GCSF-treated CCI rats exhibited significantly lower MCP-1 expression in the DRGS on the 7th day after the nerve injury compared to respective vehicle-treated rats. Authors assumed that an early GCSF treatment can suppress MCP-1 expressions, probably through upregulating microRNA-122 expressions in the DRGs of CCI rats at an earlier stage, thus indirectly attenuating neuropathic pain development.

This manuscript is interesting and provides new insights and knowledge about the effect of the GCSF (treatment) in the DRGs of CCI rats at  very early stages (day 1 and day 7 post constriction injury).

Nevertheless, this manuscript needs corrections and improvements before publishing is possible.

General points:

  • Please add a list of abbreviations.
  • Please check and correct in the whole manuscript all spaces between the text and numbers of the citied references.

Special points:

  • Keywords: Please add also to keywords: dorsal root ganglia; rats

  • Introduction

The introduction section has to improved by adding a schematic time line, where the reader could exactly see, on which day what happened to the animals and which analyses were when performed.

Lines 51-68: please describe more exactly all studies cited in all these sentences. 

Line 68: please describe more exactly the NanoString nCounter Analysis System.

Line 75: please describe more exactly the microRNA-122. Let the reader get some knowledge on that topic.

  • Materials and Methods

Line 80: you said: Adult male Sprague–Dawley rats (BioLASCO Taiwan Co., Ltd., Taipei, Taiwan). Please add the total number of the rats used in all your experiments in this study. Please also say, how many rats were evaluated in each special investigated topic.

Lines 82-85: you said: All  procedures were conducted in accordance with the Guidelines for Care and Use of Laboratory Animals and were approved by the Institutional Animal Care and Use Committee (IACUC) at the Chang Gung Memorial Hospital (IACUC: 2018120401).

Please add the exactly date of the permission of your experiments.

Lines 91- 94: You said: Muscle dissection without manipulation of the right sciatic nerve was performed in the sham group. A single dose of GCSF (200 μg/kg, Filgrastim; Kyowa Hakko Kirin, Japan) was injected intravenously (i.v.) immediately after surgery. The same amount of normal saline was injected into the vehicle control groups.

What do you mean with “muscle dissection”? Please describe exactly.

Please add also the number of the animals in each group: experimental group, sham group.

Line 97: you said: Each animal was placed in a 30 × 30 × 15 cm transparent box for a 10-min habituation period. What about this box? Why did you put the animals before testing in a transparent box? Please add your comments. Please give literature in that respect. Give  2-3 sentences on the testing with Frey hairs (in a box, in rats in hands by a second experimenter, ..)

Lines 96-103: please add the number of the animals in each group used in behavioral tests for mechanical allodynia.  

Lines 106-107: you said: The right L5 and L6 dorsal root ganglia (DRGs) were separated and collected in liquid nitrogen. Why did you only take specifically the L5 und L6 dorsal root ganglia on the experimental side – and not also on the contralateral side or above or below the L5,6 for comparison and prove of principle?

Lines 117-118: please add references at the end of this sentence.

Lines 119-120: please add more information about Panel Code Set.

Lines 112-128: please add the number of the animals in each group used for nCounter data analysis.

Lines129-135:  please add the number of the animals in each group used for ELISA.

Lines 137-138: you said: The rats were deeply anesthetized with sodium pentobarbital and transcardially perfused with PBS (Sigma, USA) followed by a fixative solution containing 4% paraformaldehyde.  

Please add the number of the animals in each group used for immunohistochemistry.

Lines 155-157: you said: The numbers of MCP-1 + CGRP-positive, MCP-1 + IB4-positive, and MCP-1 + NF200-positive neurons in the L4-L6 DRGs of the different groups were counted by an investigator who was blinded to the status of the animals manually.

Important: please describe exactly how the numbers of MCP-1 + CGRP-positive, MCP-1 + IB4-positive, and MCP-1 + NF200-positive neurons in the L4-L6 DRGs of the different groups were counted by the investigator who was blinded to the status of the animals manually?  Here the method used is extremely important for your message.

  • Results

Figure 1: for better understanding and legibility please used instead of different lines pattern a different colours for the different groups line patters.

Figures are quite nice and of high quality.

  • Discussion

The Discussion section should be partially re-written: please draw your Discussion section in the same way and use the same sections titles as in your Results section and discuss like this all your results.  

Lines 288-291: please add references at the end of this sentence: Many studies have investigated the temporal profiles of microRNA expression in the DRGs of CCI rats, but those studies usually focused on microRNA expression in the late stage after nerve injury and did not achieve consistent results.

Lines 317-318: please add references at the end of this sentence: From previous studies, we proposed that the small amount of microRNA-122 was probably conveyed to the DRGs by the bloodstream. 

Lines 324-326: please add references at the end of this sentence: MCP-1 is a chemokine that regulates the migration and infiltration of monocytes/macrophages in inflammatory conditions and has an important role in neuropathic pain development. 

Author Response

Response to reviewer 5:

Thanks for your constructive comments and detailed and extensive suggestions. We have added a list of abbreviations at the end of manuscript. We have checked and corrected in the whole manuscript all spaces between the text and numbers of the citied references. We have added “dorsal root ganglia; rats” to the keywords as reviewer 5 suggested.

In the Introduction:

We have added a schematic timeline (Figure 1) to help our readers to understand our previous studies.

  • lines 54 to 65: We have added more detailed descriptions about our previous studies.
  • lines 75 to 76, and lines 150 to 158: We have added more detailed descriptions about NanoString nCounter Analysis System.
  • lines 81 to 87: We have added more detailed descriptions about the function and role of microRNA-122 in neurobiology, which you will see in the revised introduction.

In the Materials and Methods

  • lines 104-105: We have added the total number of the rats used in all experiments in this study. On day 1 after surgery, total number of rats was 33. We used three rats in each group for microRNAs studies, and eight rats in each group for ELISA studies (n = 4 in IL-1β and MCP-1. We increased rat number to 8 in each group in IL-6 study and expected that there will be a statistically significant difference between each group. However, the differences between the groups did not reach statistical significances). On day 7 after surgery, total number of rats was 42. We used two rats in each group for microRNAs studies, eight rats in each group for ELISA studies, and four rats in each group for IHC studies.
  • line 107-108: We have added the exact date of the IACUC permission of our experiments. (December 24, 2018)
  • lines 112-113: We have replaced the words “muscle dissection” to more exactly descriptions.” Muscles on the right inguinal region were separated by forceps to let sciatic nerve be exposed.”, which were performed in both sham and experimental groups.
  • lines 119-121: We have added the number of the animals in each group as reviewer 5 suggested.
  • lines 124-126: We have added the references and description about the transparent box for the habituation period. The transparent box for the habituation period was used to reduce stress of rats and avoid the errors of behavioral tests.
  • lines 129-131: We have described the Von Frey hairs tests in detail. “When the rats showed a sharp withdrawal response or a flinch to the given filament, the bending force of that filament was defined as the mechanical threshold intensity. When a withdrawal response was established, a filament with the next lower force was used and restarted the ascending order. The final hind paw withdrawal threshold was defined as the lowest force that caused at least three withdrawals out of five consecutive applications”.
  • lines 138-139: The right L5 and L6 dorsal root ganglia (DRGs) were separated. We also collected the left L5/6 DRGs at the same time. Left DRGs were not studied at the current study. However, our previous study [1] had evaluated TNF-α levels on bilateral DRGs of CCI rats with GCSF treatment. Only the right DRGs (injured site) of CCI rats showed a significantly increased TNF-α level compared to the contralateral (left) side and naïve group, suggesting that only the DRGs in the injured side show a significant increase of neuroinflammation.
  • lines 150: We have added the reference at the end of this sentence.
  • lines 150-158: We have added more information about Panel Code Set. “Briefly, unique DNA nucleic acid tags for each microRNAs species were ligated to the 3’ end of each mature microRNA. Each unique nucleic acid tags are linked to special Panel Code set coded by several different sequential fluorescent molecules. Then, the specifically tagged microRNAs were counted via hybridization with the nCounter microRNA Panel Code Set. Each barcode with special sequential fluorescence molecules was collected and counted on the nCounter Digital Analyzer (NanoString Technologies) from individual fluorescent barcodes and quantified each microRNAs molecule present in each sample. Basically, one barcode with special sequential fluorescence read by machine represents one special microRNA molecule”.
  • lines 163-164: We have added the number of the animals in each group used for nCounter data analysis.
  • line 171: We have added the number of the animals in each group used for ELISA.
  • lines 179-180: We have added the number of the animals in each group used for immunohistochemistry.
  • lines 194-195: We have described how to count the numbers of double-stained neurons in detail. Basically, we counted the numbers of positive double-stained neurons in the whole field of each slide.

In the Results

  • Figure 1: we have used different colors to represent different experiment group.

In the Discussion

We have reorganized our discussion section in the same way as in our results section. We have used subtitles in each paragraph.

  • lines 330-333: We have added the references at the end of the sentence. “Many studies have investigated the temporal profiles of microRNA expression in the DRGs of CCI rats, but those studies usually focused on microRNA expression in the late stage after nerve injury and did not achieve consistent results [18-21,24, 25]”.
  • lines 362-363: We have added the references at the sentence. “From previous studies [34-36], we proposed that the small amount of microRNA-122 was conveyed to the DRGs by the bloodstream”.
  • lines 370-372: We have added the references at the sentence. “MCP-1 is a chemokine that regulates the migration and infiltration of monocytes/macrophages in inflammatory conditions and has an important role in neuropathic pain development [39-43]”.

Reference:

  1. Chao, P.K.; Lu, K.T.; Lee, Y.L.; Chen, J.C.; Wang, H.L.; Yang, Y.L.; Cheng, M.Y.; Liao, M.F.; Ro, L.S. Early systemic granulocyte-colony stimulating factor treatment attenuates neuropathic pain after peripheral nerve injury. PLoS One 2012, 7, e43680.

Round 2

Reviewer 1 Report

Most of my concerns have been addressed in the revised version; and therefore, I have no further comment on this manuscript.

Author Response

Thanks for your positive and favorable comments. We have added your constructive comments and suggestions in our revised manuscript.

Reviewer 5 Report

This manuscript was impressively improved and corrected according to almost everyone proposal I had in the first draft.

Nevertheless, this manuscript need some additional correction or clarifications before publishing is possible:

Special points:

Introduction

You said: We have added a schematic timeline (Figure 1) to help our readers to understand our previous studies.

This new Figure 1 is a fine idea, but this Figure has a very bad quality and is absolutely unreadable.

Please check or correct Figure 1. (Possibly, I did not get sharp versions of all figures)

Materials and Methods

Line 146: you said: The rats were anesthetized with sodium pentobarbital and transcardially perfused with PBS (Sigma, USA) on the 1st and 7th days after surgery.

Please add the dose of pentobarbital. 

Lines 147-148: you said: The right L5 and L6 dorsal root ganglia (DRGs) were separated and collected in liquid nitrogen.

My previously question was: Why did you took specifically the right sided L5 und L6 dorsal root ganglia only?

You answer: The right L5 and L6 dorsal root ganglia (DRGs) were separated. We also collected the left L5/6 DRGs at the same time. Left DRGs were not studied at the current study. However, our previous study [1] had evaluated TNF-α levels on bilateral DRGs of CCI rats with GCSF treatment. Only the right DRGs (injured site) of CCI rats showed a significantly increased TNF-α level compared to the contralateral (left) side and naïve group, suggesting that only the DRGs in the injured side show a significant increase of neuroinflammation.

Please add all your comment to the Discussion section.

Lines 170-173: my previously question was: please add the number of the animals in each group used for nCounter data analysis.

Your comment: The statistical differences of normalized data of different microRNAs expressions between each group were further analyzed by the default setting of  NanoString nSolver software 3.0. (n = 3 and 2 in each group on the 1st and 7th days after surgery, respectively)

The number of the animals in each group used for nCounter data analysis is very low: only n=3 and 2 in each group. Please add your comments once again.    

Lines 207-210: you said: The numbers of MCP-1 + CGRP-positive, MCP-1 + IB4-positive, and MCP-1 + NF200-positive neurons in the L4-L6 DRGs of the different groups were counted by an investigator who was blinded to the status of the animals manually.

My previously question was: please describe exactly how the numbers of MCP-1 + CGRP-positive, MCP-1 + IB4-positive, and MCP-1 + NF200-positive neurons in the L4-L6 DRGs of the different groups were counted by an investigator who was blinded to the status of the animals manually?   

Your new comment: The investigator manually counted the numbers of positive double-stained neurons in the whole field of each slide.

I think is it not possible to count exactly the number of positive stained neurons in the whole field of each slide? And then, did you use some software to analyse the cell numbers?

Results

Lines 246-262: you said: 3.2. GCSF upregulated microRNA-122 expression in the DRGs of CCI rats on the 1st day after nerve injury The levels of microRNA-7b, microRNA-19a, microRNA-122, and microRNA-598-3p were significantly decreased, but the levels of microRNA-141 were significantly increased in the DRGs of the vehicle-treated CCI rats compared to the sham controls on the 1st day after nerve injury.

However, GCSF treatment only reversed the levels of microRNA-122 expression in the DRGs of CCI rats on the 1st day after nerve injury. The levels of microRNA-122 in the DRGs of the GCSF-treated CCI rats were significantly higher than those in the vehicle-treated CCI rats. In contrast, the levels of microRNA-7b, microRNA-19a, microRNA-141, and microRNA-598-3p exhibited similar expression in the DRGs of the vehicle-treated and GCSF-treated CCI rats (Supplement 3). The other screened microRNAs also did not show a similar trend as that of microRNA-122 on the 1st day after nerve injury. However, there were no significant differences in microRNA-122 levels in the DRGs between the different groups on the 7th day after nerve injury (Day 1: n = 3 in each group; Day 7: n = 2 in each group; #P < 0.05: vehicle-treated rats compared to sham-operated rats; *P < 0.05: GCSF-treated CCI

Also, the same in the appropriate Figure 3: you said: the other screened microRNAs did not show a similar trend. The data are shown as the means ± SEMs (Day 1: n = 3 in each group; Day 7: n = 2 in each group; #P < 0.05: vehicle-treated rats compared to sham-operated rats; *P < 0.05: GCSF-treated CCI rats compared to vehicle-treated CCI rats, unpaired t test between each group by the default setting of NanoString nSolver software 3.0). 

I think, only 3 or 2 animals in each group is very low for this results interpretation, and also fewer than for other experiments carried out in your study. Please give a comments.

Author Response

In the Introduction

You said: We have added a schematic timeline (Figure 1) to help our readers to understand our previous studies. This new Figure 1 is a fine idea, but this Figure has a very bad quality and is absolutely unreadable.

Please check or correct Figure 1. (Possibly, I did not get sharp versions of all figures)

Response:

Thanks for your constructive comments and meticulous suggestions. We have modified the Figure 1 and increased the size of all characters. We hoped that this new version will be sharper and clearer to be read.

In the Materials and Methods

Line 146: you said: The rats were anesthetized with sodium pentobarbital and transcardially perfused with PBS (Sigma, USA) on the 1st and 7th days after surgery.

Please add the dose of pentobarbital.

Response:

Line 146 (revised lines 137 and 174): We have added descriptions “The rats were anesthetized with sodium pentobarbital (50 mg/kg body weight) and transcardially perfused with PBS (Sigma, USA) on the 1st and 7th days after surgery.”

Lines 147-148: you said: The right L5 and L6 dorsal root ganglia (DRGs) were separated and collected in liquid nitrogen.

My previously question was: Why did you took specifically the right sided L5 und L6 dorsal root ganglia only?

You answer: The right L5 and L6 dorsal root ganglia (DRGs) were separated. We also collected the left L5/6 DRGs at the same time. Left DRGs were not studied at the current study. However, our previous study had evaluated TNF-α levels on bilateral DRGs of CCI rats with GCSF treatment. Only the right DRGs (injured site) of CCI rats showed a significantly increased TNF-α level compared to the contralateral (left) side and naïve group, suggesting that only the DRGs in the injured side show a significant increase of neuroinflammation.

Please add all your comment to the Discussion section.

Response:

Thanks reviewer’s suggestions. We have added the comments “Our previous study had evaluated TNF-α levels on bilateral DRGs of CCI rats with GCSF treatment [1]. Only the right DRGs (injured site) of CCI rats showed a significantly increased TNF-α level compared to the contralateral (left) side and naïve group, suggesting that only the DRGs in the injured side show a significant increase of neuroinflammation. Thus, the MCP-1 expression on the right L5 and L6 DRGs were studied but left DRGs were not studied in the current study.” to the discussion section (revised lines 393 to 398).

Lines 170-173: my previously question was: please add the number of the animals in each group used for nCounter data analysis.

Response:

Thanks for reviewer’s suggestions. We have added the number of the animals in each group used for nCounter data analysis in the revised manuscript in lines 164-167.

Your comment: The statistical differences of normalized data of different microRNAs expressions between each group were further analyzed by the default setting of  NanoString nSolver software 3.0. (n = 3 and 2 in each group on the 1st and 7th days after surgery, respectively)

The number of the animals in each group used for nCounter data analysis is very low: only n=3 and 2 in each group. Please add your comments once again.

Response:

Although the number of the animals in each group for nCounter data analysis is quite low, the statistical differences of normalized data of different microRNAs expressions between each group were analyzed by the default setting of NanoString nSolver software 3.0. (n = 3 in each group on the 1st day after surgery), which are reaching statistically significant. Our Institutional Animal Care and Use Committee (IACUC) required animals’ studies in our hospital to sacrifice as less animals as possible, thus we ceased sacrificing more rats in this study. Nevertheless, we agreed that further studies of microRNA-122 mimics/inhibitors to confirm the analgesic effect of microRNA-122 are warranted (Lines 416-417).

Lines 207-210: you said: The numbers of MCP-1 + CGRP-positive, MCP-1 + IB4-positive, and MCP-1 + NF200-positive neurons in the L4-L6 DRGs of the different groups were counted by an investigator who was blinded to the status of the animals manually.

My previously question was: please describe exactly how the numbers of MCP-1 + CGRP-positive, MCP-1 + IB4-positive, and MCP-1 + NF200-positive neurons in the L4-L6 DRGs of the different groups were counted by an investigator who was blinded to the status of the animals manually?  

Your new comment: The investigator manually counted the numbers of positive double-stained neurons in the whole field of each slide.

I think is it not possible to count exactly the number of positive stained neurons in the whole field of each slide? And then, did you use some software to analyze the cell numbers?

Response:

Thanks for reviewer’s expert and professional comments. We have used software “MetaMorph (version 7.8; Molecular Devices, USA)” to count the number of positive double-stained neurons. Unfortunately, the methods did not get satisfactory results because of the background interference. The number of positive double-stained neurons is not much (less than 30 in each slide). Thus, we could count the number of positive double-stained neurons in the whole field of each slide (every fourth section was picked from a series of consecutive DRGs [10 μm]) manually, which showed quite consistent and less intra-variable results (Lines 197-199).

In the Results

Lines 246-262: you said: GCSF upregulated microRNA-122 expression in the DRGs of CCI rats on the 1st day after nerve injury The levels of microRNA-7b, microRNA-19a, microRNA-122, and microRNA-598-3p were significantly decreased, but the levels of microRNA-141 were significantly increased in the DRGs of the vehicle-treated CCI rats compared to the sham controls on the 1st day after nerve injury.

However, GCSF treatment only reversed the levels of microRNA-122 expression in the DRGs of CCI rats on the 1st day after nerve injury. The levels of microRNA-122 in the DRGs of the GCSF-treated CCI rats were significantly higher than those in the vehicle-treated CCI rats. In contrast, the levels of microRNA-7b, microRNA-19a, microRNA-141, and microRNA-598-3p exhibited similar expression in the DRGs of the vehicle-treated and GCSF-treated CCI rats (Supplement 3). The other screened microRNAs also did not show a similar trend as that of microRNA-122 on the 1st day after nerve injury. However, there were no significant differences in microRNA-122 levels in the DRGs between the different groups on the 7th day after nerve injury (Day 1: n = 3 in each group; Day 7: n = 2 in each group; #P < 0.05: vehicle-treated rats compared to sham-operated rats; *P < 0.05: GCSF-treated CCI

Also, the same in the appropriate Figure 3: you said: the other screened microRNAs did not show a similar trend. The data are shown as the means ± SEMs (Day 1: n = 3 in each group; Day 7: n = 2 in each group; #P < 0.05: vehicle-treated rats compared to sham-operated rats; *P < 0.05: GCSF-treated CCI rats compared to vehicle-treated CCI rats, unpaired t test between each group by the default setting of NanoString nSolver software 3.0).

I think, only 3 or 2 animals in each group is very low for this results interpretation, and also fewer than for other experiments carried out in your study. Please give a comment.

Response:

Although the number of the animals in each group for nCounter data analysis is quite low, the differences between each group on the 1st day are reaching statistically significant. We have consulted the technical specialists of Cold Spring Biotech Corp (The Representative in Taiwan), and they gave us the conclusion that statistically significant differences between each group on the 1st day are quite constant and firmly confirmed, suggesting that we did not need to sacrifice more rats to re-confirm the results. Moreover, we have also checked the results by correlating the microRNA-122 related chemokines/cytokines (MCP-1) studies, which are consistent with our hypotheses. Nevertheless, further studies of microRNA-122 mimics/inhibitors to confirm the analgesic effect of microRNA-122 are warranted (Lines 416-417).